# Sources of Sleep Disturbances and Psychological Strain for Hospital Staff Working during the COVID-19 Pandemic

**DOI:** 10.3390/ijerph18126289

**Published:** 2021-06-10

**Authors:** Nasrin Abdoli, Vahid Farnia, Somayeh Jahangiri, Farnaz Radmehr, Mostafa Alikhani, Pegah Abdoli, Omran Davarinejad, Kenneth M. Dürsteler, Annette Beatrix Brühl, Dena Sadeghi-Bahmani, Serge Brand

**Affiliations:** 1Substance Abuse Prevention Research Center, Health Institute, Kermanshah University of Medical Sciences, Kermanshah 6719851115, Iran; Abdolinasrin511@yahoo.com (N.A.); vahidfarnia@yahoo.com (V.F.); Jahangirisomayeh84@gmail.com (S.J.); m.alikhani18@yahoo.com (M.A.); pegahabdoli@yahoo.com (P.A.); dena.sadeghibahmani@upk.ch (D.S.-B.); 2Clinical Research Development Center, Imam Khomeini and Mohammad Kermanshahi and Farabi Hospitals, Kermanshah University of Medical Sciences, Kermanshah 6719851451, Iran; radmehr.f12@gmail.com (F.R.); odavarinejad@gmail.com (O.D.); 3Psychiatric Clinics, Division of Substance Use Disorders, University of Basel, 4002 Basel, Switzerland; Kenneth.Duersteler@upk.ch; 4Center for Addictive Disorders, Department of Psychiatry, Psychotherapy and Psychosomatics, Psychiatric Hospital, University of Zurich, 8001 Zurich, Switzerland; 5Center for Affective, Stress and Sleep Disorders (ZASS), Psychiatric University Hospital Basel, 4002 Basel, Switzerland; annette.bruehl@upk.ch; 6Sleep Disorders Research Center, Kermanshah University of Medical Sciences, Kermanshah 6719851451, Iran; 7Departments of Physical Therapy, University of Alabama at Birmingham, Birmingham, AL 35209, USA; 8Department of Sport, Exercise and Health, Division of Sport Science and Psychosocial Health, University of Basel, 4052 Basel, Switzerland; 9School of Medicine, Tehran University of Medical Sciences, Tehran 1417466191, Iran

**Keywords:** hospital staff members, COVID-19, depression, anxiety, stress, self-efficacy, social support, insomnia

## Abstract

Hospital staff members reported increased stress-related workload when caring for inpatients with COVID-19 (“frontline hospital staff members”). Here, we tested if depression, anxiety, and stress were associated with poor sleep and lower general health, and if social support mediated these associations. Furthermore, we compared current insomnia scores and general health scores with normative data. A total of 321 full-time frontline hospital staff members (mean age: 36.86; 58% females) took part in the study during the COVID-19 pandemic. They completed a series of questionnaires covering demographic and work-related information, symptoms of depression, anxiety, stress, social support, self-efficacy, and symptoms of insomnia and general health. Higher symptoms of depression, anxiety, and stress were associated with higher symptoms of insomnia and lower general health. Higher scores of depression, anxiety, and stress directly predicted higher insomnia scores and lower general health scores, while the indirect effect of social support was modest. Compared to normative data, full-time frontline hospital staff members had a 3.14 higher chance to complain about insomnia and a significantly lower general health. Symptoms of insomnia and general health were unrelated to age, job experience, educational level, and gender. Given this background, it appears that the working context had a lower impact on individuals’ well-being compared to individual characteristics.

## 1. Introduction

Like all countries, Iran is struggling with the COVID-19 pandemic, which is a challenge for societies, individuals, and healthcare systems [1,2]. Relatedly, to decrease the spread of the virus and prevent further deaths and severe cases of infection, the national government, advised by health authorities, imposed confinements [3,4,5]. To illustrate this, health authorities temporarily legislated to close borders, schools, universities, sports events, and religious and cultural centers, and to disallow gatherings in open spaces.

Furthermore, by 14 January 2021, in Iran, about 16,291 hospital beds were needed to treat COVID-19 patients, while about 45,159 hospital beds were available (https://covid19.healthdata.org/iran-(islamic-republic-of)?view=resource-use&tab=trend&resource=all_resources (Retrieved on 14 January 2021)).

Despite such favorable numbers of available hospital beds, data from rapid reviews and meta-analyses showed that hospital staff members in contact with affected patients (“frontline hospital staff members”) appeared to be at an increased risk to report symptoms of exhaustion, depression, and anxiety [6]. In general, health care workers were at an increased risk of developing short- and long-term mental health problems [7], and compared to non-frontline hospital staff members, frontline hospital staff members reported higher levels of acute and post-traumatic stress symptoms and psychological distress [6,8]. In particular, the following risk factors were reported: being younger, being more junior, being the parents of dependent children, having an infected family member, lack of practical support, and stigma [6]. Moreover, a rapid review showed that depression, anxiety, and psychological distress were common among hospital staff members in contact with infected patients [9]. The use of masks and general infection control training were associated with decreased infection risk, while the impact of such devices on psychological health was not reported.

Concerning the psychological distress of hospital staff members in Iran, frontline nurses reported higher scores of job-related stress and burnout when compared to non-frontline nurses [10]; age and job experience were not confounders, and social support of friend and families were equal in both groups. By contrast, higher age, higher educational degrees, and being male predicted lower stress among frontline nurses aged 40.6 years on average [11]. Among 495 nurses (71% females), job-related mental pressure, physical pressure, and time pressure were associated with being female, being a frontline nurse, and reporting a lower educational/vocational level [12]. Importantly, a higher task load was associated with lower general health [12]. Among 869 nurses (71.2% females), higher prevalence rates of depression (41.7%), anxiety (52.2%), and stress (53.9%) were related to female gender and being a frontline hospital staff member [13]. Results from quantitative studies showed that frontline nurses reported anxieties related to the disease, fear of infecting their families, emotional distress of delivering bad news, and conflicts between (dysfunctional) fears and the conscience to accomplish an important job [14]. Other frontline nurses appreciated supportive resources and social support [15]. Among 599 health care practitioners working in eight different Iranian cities, quality of life was associated with higher social support [16]. About 35% of 761 (frontline) nurses reported fear of getting infected, while the subjective feeling of self-efficacy was associated with protective behavior [17].

For sleep complaints in general, and insomnia in particular, it appears that no study has focused on insomnia among frontline hospital staff members in Iran. This is surprising for several reasons: Following the hyperarousal hypothesis of insomnia [18], psychophysiological arousal leads to insomnia at the short and long term. Psychiatric issues and poor sleep were related [19,20], as also poor sleep, poor emotion regulation, and stress were related [21,22,23,24]. Salari et al. [25] summarized in their systematic review and meta-analysis of seven studies that stress was the major cause of sleep disturbances among medical staff members and nurses during the COVID-19 pandemic. Given this background and given the lack of data among frontline hospital staff members in Iran, the key aim of the present study was to investigate the associations between psychological issues (depression, anxiety, stress) and insomnia. We also examined the associations between psychological issues and general health, while social support was further introduced as a mediating factor. To this end, 321 frontline hospital staff nurses completed a series of questionnaires on depression, anxiety, stress, self-efficacy, social support, insomnia, and general health.

The following two hypotheses and one research question were formulated. First, following others [18,21,22,25,26], we expected that higher scores of depression, anxiety, and stress would be related to insomnia and lower general health. Second, following others [11,12,13], we expected that frontline hospital staff members would report higher insomnia and lower general health, compared to normative data of healthy people. In doing so, we would be able to judge if hospital staff members reported similar or higher insomnia scores when compared to the general population. The exploratory research question was: Does social support mediate the association between depression, anxiety, stress, insomnia, and general health? The question was exploratory, because social support was either an important health factor [15,16] or not [10]. We suggest that the pattern of results might help to identify if and which frontline hospital staff members might need particular support to cope with COVID-19-related psychological issues.

## 2. Methods

### 2.1. Study Procedure

Frontline and full-time working hospital staff members of the Farabi Hospital (Kermanshah University of Medical Sciences, Kermanshah, Iran) were approached between August 2020 and November 2020 to participate in this cross-sectional study on psychological issues (depression, anxiety, stress), self-efficacy, social support, insomnia, and general health. To this end, the study was posted on the intranet and social network sites of the hospital. Staff members interested in participation could click and follow the link to be contacted by a study team member. All participants were informed about the aims of the study and that their data would be anonymous and held securely. Thereafter, they signed a written informed consent. Participants completed a series of questionnaires covering sociodemographic data, educational/vocational training, job duration, and depression, anxiety, stress, self-efficacy, social support, insomnia, and general health. Participants needed about 20 min to complete all questionnaires. The ethical committee of the Kermanshah University of Medical Sciences (KUMS; Kermanshah, Iran: register nr: 3010523) approved the study, which was performed in accordance with the current revision [27] of the Declaration of Helsinki.

### 2.2. Participants

Of the 730 staff members of the hospital, a total of 588 were working frontline. Of those, the research team contacted 429 individuals. Inclusion criteria were: 1. Age between 18 and 65 years; 2. working full time as a hospital staff member; 3. working as frontline staff during the entire time of the COVID-19 pandemic; 4. self-reporting either early or normal daytime shift within the last six months with routine day-offs; 5. signed written informed consent. Exclusion criteria were: 1. Irregular shifts, including night shifts within the last six months; 2. unable or unwilling to comply with the study requirements (e.g., completing self-rating questionnaires in Farsi); 3. self-reporting sleep disturbances as a result of caring for small children (including breast feeding) or obstructive sleep apnea; 4. intake of sleep-altering medications such as antihistaminic medication or prescribed psychopharmaceutic medication. Of the 429 initially approached, 29 were not working full time, or were not frontline staff, or working night shifts. Next, of the 400 persons initially interested in the study, 321 (80.25%) agreed to participate. Table 1 provides participants’ descriptive and inferential characteristics, both for the whole sample and separately for female and male participants (see Results section).

### 2.3. Measures

#### 2.3.1. Sociodemographic and Job-Related Questions

Participants reported on their age (years), gender (female, male), civil status (single, married, divorced); duration of job experience (<1, 1–2, 3–5, 6–10, >11 years), and educational background (high school and diploma, high diploma, bachelor degree, master’s degree).

#### 2.3.2. Depression, Anxiety, and Stress (DASS-21)

To assess depression, anxiety, and stress, the DASS-21 was employed [28]. The Farsi version showed satisfactory psychometric properties [29]. This questionnaire consists of 21 questions on depression (7 questions), anxiety (7 questions), and stress (7 questions). Answers are given on four-point rating scales ranging from 0 (= does not apply to me at all) to 3 (= absolutely applies to me), with higher sum scores reflecting higher depression, anxiety, and stress (Cronbach’s alpha: 0.89).

#### 2.3.3. Social Support—Multidimensional Scale of Social Support (MSPSS)

To assess social support, participants completed the Multidimensional Scale of Social Support (MSPSS) [30,31]. The Farsi version showed satisfactory psychometric properties [32]. Subscales focus on social support from family, friends, and significant others in life. Answers are given on a seven-point Likert scales ranging from 0 (=strongly disagree) to 6 (=strongly agree), with higher sum scores reflecting a higher subjectively perceived social support (Cronbach’s alpha: 0.87).

#### 2.3.4. Self-Efficacy—General Self-Efficacy Scale-10 (GES-10)

To assess self-efficacy, participants completed the 10-item General Self-Efficacy Scale-10 [33]. The Farsi version showed satisfactory psychometric properties [34]. Answers are given on a four-point Likert scale with the anchor points 1 (=not correct at all) to 4 (=completely correct). Sum scores range between 10 and 40, with higher scores reflecting higher self-efficacy (Cronbach’s alpha: 0.90).

#### 2.3.5. Insomnia—Athens Insomnia Scale (AIS)

The Athens Insomnia Scale (AIS) [35] comprises eight items used to assess the severity and effects of insomnia. The Farsi version had satisfactory psychometric properties [36]. All items are rated using a four-point Likert scale (0 = no problem or equivalent meaning; 3 = very severe problem or equivalent meaning). The total sum score ranges from 0 to 24, with higher scores reflecting higher self-rated insomnia. The cut-off point is 6 [35] (Cronbach’s alpha: 0.92).

#### 2.3.6. General Health

The General Health Questionnaire-12 (GHQ-12) is a measure used to self-assess current mental health [37]. The Farsi version showed satisfactory psychometric properties [38]. Answers are given on four-point Likert scales with the anchor points 0 (=not at all) to 3 (=definitely true), with higher mean scores reflecting a higher general health (Cronbach’s alpha: 0.91).

### 2.4. Statistical Analysis

Preliminary calculations: With a series of Spearman rank correlations, the associations between age, duration of job experience, and educational level and dimensions of depression, anxiety, stress, self-efficacy, social support, general health, and insomnia were calculated. Correlation coefficients were between rho = −0.03 and 0.04; it followed that age, duration of job experience, and educational level were not introduced as possible confounders. Next, with a series of t-tests, gender differences for depression, stress, anxiety, self-efficacy, social support, general health, and insomnia were tested; t-values were between 0.10 and 0.80; it followed that gender was not introduced as a possible confounder.

An X^2^-test was performed to compare insomnia scores (yes vs. no) with normative data taken from Morin et al. [39].

A one-sample t-test was performed to compare participants’ general health mean score with the mean of score (3.70) of a normative sample of healthy adults [40].

A series of Pearson’s correlations was performed to calculate the associations between dimensions of depression, anxiety, stress, self-efficacy, social support, insomnia, and general health.

Next, two models were performed to test the direct and indirect effects of depression, anxiety, stress and social support on insomnia and general health scores. To this end, in the first model, depression, anxiety, and stress predicted directly insomnia and general health, and indirectly via social support. In the second model, social support directly predicted insomnia and general health, and indirectly via depression, anxiety, and stress.

Preliminary conditions to perform multiple regression analyses were met [41,42,43]: N = 321 > 100; predictors explained the dependent variables (Rs = 0.506–0.539, R^2^s = 0.26–0.29; the number of predictors (4) · 10 = 40 < N (321)), and the Durbin–Watson coefficient was between 1.5 and 2.5, indicating that the residuals of the predictors were independent. Furthermore, the variances inflation factors (VIF) were between 1.20 and 1.50; while there are no strict cut-off points to report the risk of multicollinearity, VIF < 1 and VIF > 10 indicate multicollinearity [41,42,43].

All statistical computations were performed with SPSS^®^ 25.0 (IBM Corporation, Armonk, NY, USA) for Apple Mac^®^.

## 3. Results

### 3.1. Preliminary Calculations

With a series of Spearman rank correlations the associations between age, duration of job experience, and educational level and dimensions of depression, anxiety, stress, self-efficacy, social support, general health, and insomnia were calculated. Correlation coefficients were between rho = −0.03 and 0.04; it followed that age, duration of job experience, and educational level were not introduced as possible confounders. Next, with a series of t-tests, gender differences for depression, stress, anxiety, self-efficacy, social support, general health, and insomnia were tested; t-values were between 0.10 and 0.80; it followed that gender was not introduced as possible confounder.

### 3.2. Participants’ Characteristics

Table 1 provides the descriptive and statistical overview of participants’ sociodemographic and work-related characteristics. Table 1 also reports the descriptive statistics of depression, anxiety, stress, self-efficacy, social support, insomnia, and general health.

### 3.3. Correlation Coefficients between Depression, Anxiety, Stress, and Self-Efficacy Dimensions of Social Support, Insomnia and General Health

Table 2 provides the correlation coefficients between dimensions of depression, anxiety, stress, self-efficacy, of social support, insomnia, and general health.

Scores of depression, anxiety and stress were highly interrelated (large correlations). Such higher scores were also associated with dimensions of lower social support (medium correlations) and lower general health (medium to large correlations), and higher dimensions of insomnia (medium to large correlations). Depression, anxiety, and stress were unrelated to self-efficacy (trivial correlations).

Higher scores of self-efficacy were related to higher social support (small correlations) and higher general health (small to medium correlations) but not to insomnia (trivial correlation).

Higher social support was related to higher general health (medium correlations) and lower insomnia (medium correlations), whereas only negative symptoms of general health were associated with insomnia (small correlation).

### 3.4. Insomnia Scores; Dichotomization: Yes vs. No, and Comparison with Normative Data

For the Athens Insomnia Scale (AIS), following Soldatos et al. [35], a score of six and more points indicates insomnia. Of the 321 participants, 44 (13.7%) had an insomnia score of 0 to 5 points; 272 (86.3%) had a score of six points and higher. Following Morin et al. [39], 37.5% of the general population report insomnia. Compared to the general population, participants reported statistically significantly higher insomnia (X^2^(N = 642, df = 1) = 201.32, *p* = 0.0001). The odds to report insomnia was 3.14-fold higher (CI: 2.64–3.75) among participants compared to the general population.

### 3.5. Mean Score of the General Health Questionnaire

For the General Health Questionnaire-12, Montazeri et al. [40] reported a mean score of 3.70 among healthy adults. In comparison, participants had a statistically significantly lower general health mean score (M = 1.30; SD = 0.42; t(320) = 102.87, *p* = 0.000).

### 3.6. Testing the Direct and Indirect Effects of Depression, Anxiety, Stress, and Social Support on Insomnia and General Health

#### 3.6.1. Insomnia

Two models were tested (Table 3 and Table 4). In the first model, depression, anxiety, and stress predicted insomnia directly, and indirectly via social support. In the second model, social support predicted insomnia directly, and indirectly via depression, anxiety and stress.

To predict insomnia, results were as follows:

Comparing the two models, Model 1 had greater predictive power than Model 2; thus, it turned out that depression, anxiety, and stress directly predicted insomnia, whereas the indirect contribution of social support was modest.

#### 3.6.2. General Health

Two models were tested (Table 5 and Table 6). In the first model, depression, anxiety and stress predicted insomnia directly, and indirectly via social support. In the second model, social support predicted insomnia directly, and indirectly via depression, anxiety and stress.

To predict general health, results were as follows:

Comparing the two models, Model 1 had greater predictive power than Model 2; thus, it turned out that depression, anxiety, and stress directly predicted general health, whereas the indirect contribution of social support was modest.

## 4. Discussion

The key findings of the present study were that among full-time frontline hospital staff members involved in the management of patients with COVID-19, higher scores of depression, anxiety, and stress were associated with higher insomnia and lower general health. Furthermore, frontline hospital staff in the present study reported insomnia scores that were 3.14-fold higher compared to the general population, and general health was lower compared to normative data. Next, depression, anxiety, and stress predicted higher insomnia scores and lower general health scores, both directly and indirectly via lower social support, though the mediating influence of social support was modest. The present results expand upon previous research in three ways: 1. This was the first study among frontline hospital staff members in Iran to assess insomnia. 2. Scores of depression, anxiety, and stress were the main predictors of insomnia and low general health, while social support had a modest influence of insomnia and general health. 3. In this line, the contribution of demographic (age, gender) and work-related dimensions (job experience, educational level) to explain insomnia and general health was spurious. The present results are of practical importance: frontline hospital staff members appear to be at an increased risk to report psychological issues, insomnia and low general health, irrespective of their work-related and personal background.

Two hypotheses and one research question were formulated, and each of these is considered now in turn.

With the first hypothesis, we assumed that higher scores of depression, anxiety, and stress were related to insomnia and lower general health, and the data did confirm this. Accordingly, the present patterns of results comply with previous results [18,21,22,25,26]. However, the present results expand upon the current research in that this pattern of associations was observed among frontline full-time hospital staff members in Iran. Specifically, no research conducted in Iran focused so far on sleep patterns among full-time hospital staff members working with patients with COVID-19. This is surprising for the following reasons: First, restoring sleep is highly associated with favorable psychological functioning, such as low anxiety, low stress [21,44,45], optimism [46], mental toughness [47,48,49,50,51,52,53], and lower odds of suffering from psychiatric disorders [19,20]. Second, poor sleep is related to work-related issues and higher absenteeism [54,55]. Third, it appeared that hospital staff members caring for patients with COVID-19 reported particularly high scores of anxiety [9,11,14,56,57,58,59,60,61,62]; however, higher anxiety scores and poor sleep are again largely associated [21,44,45,63,64,65]. Fourth, data from studies conducted in Iran showed that frontline hospital staff members reported particularly high scores of anxiety [11,14,66,67,68]. Taken together, the results of the present study appeared to have the potential to add to the current literature in an important fashion.

With the second hypothesis we assumed that frontline hospital staff members would report higher insomnia and lower general health compared to normative data of healthy people, and again the data did confirm this. Accordingly, the results are in line with previous studies [11,12,13]. The novelty of the present findings is that we proved high insomnia and low general health among full-time frontline hospital staff compared to normative data. In doing so, it was not necessary to assess healthy controls or non-frontline full-time staff members. Furthermore, as mentioned above, we were able to show that poor sleep was associated with higher scores of anxiety, depression, and stress.

The question arises as to how such a kind of association could be explained. The following theories are offered: First, following the hyperarousal model of insomnia [18], insomnia understood as qualitatively poor sleep is the result of a bi-directional process of dysfunctional cognitive-emotional processes and neurophysiological processes, such as increased cortisol and orexin secretion and decreased adenosine secretion. Thus, poor sleep is both the consequence and the trigger of dysfunctional cognitive-emotional processes, which in turn are associated with deteriorated neurophysiological processes. Second, the cognitive-emotional hyperarousal model of Harvey and colleagues [65,69,70,71,72,73,74] assumes that dysfunctional and hyper-aroused cognitive-emotional processes unfavorably impacts on sleep onset and sleep duration. Third, the cortical hyperarousal model [75,76,77] claims that poor sleep is both the trigger and the result of an increased cortical activity. Fourth, Palmer and Alfano [78] proposed an umbrella model with poor emotion regulation as an organizing framework unfavorably impacting on sleep. Fifth, an additional theoretical and empirical direction focuses on the importance of prefrontal activity (PFC) and its impairment during sleep. Yoo et al. [79] showed that among individuals with insomnia, the PFC was less activated than limbic areas: poor sleep appeared to be the result of a decrease in top–down prefrontal control. Given this, Yoo et al. [79] speculated that a night of sleep may “reset” the correct brain reactivity to next-day emotional challenges by maintaining the functional integrity of this PFC-amygdala circuit.

To summarize, while the quality of the present results does not allow a deeper understanding of the underlying cognitive-emotional, processes, it appears plausible that participants’ poor sleep was both the trigger and the result of a lower PFC control during the night, which were associated with a hyper-aroused cortical activity, dysregulated neurophysiological processes, and dysfunctional cognitive-emotional processing.

Next, we took as exploratory the research question, if and to what extent social support mediated the association between depression, anxiety, and stress and insomnia and general health. To this end, two different models were tested. In the first model, depression, anxiety, and stress directly impacted upon insomnia and general health, and indirectly via social support. In the second model, social support directly impacted upon insomnia and general health, and indirectly via depression, anxiety, and stress. It turned out that scores of depression, anxiety, and stress had a superior impact on insomnia and general health, both directly and indirectly via social support.

With regard to the importance of social support for general psychological well-being among hospital staff members, the results are mixed: while two studies emphasized the importance of social support as a health and protective factor among frontline hospital staff members in Iran [15,16], this was not confirmed in another study [10]. Thus, on a continuum of the importance of social support ranging from no importance [10] to high importance [15,16], our results are headed towards the pole of no importance [10]. The quality of the studies does not allow a deeper understanding and comparison between the previous and current study. Given this, a reasonable explanation remains difficult.

We also note that sociodemographic (age, gender) and work-related dimensions (working experience, educational/vocational levels) were completely unrelated to psychological functioning (depression, anxiety, stress, insomnia, general health). Again, these results are at odds with previous studies: compared to their male counterparts, female frontline hospital staff members reported higher stress [11,12,13]. Furthermore, higher age and higher vocational levels were associated with lower stress [11,12]. Given these results, our findings are in contrast to previous findings [80]. As such, it appears that work-related conditions had a higher impact on the individual psychological functioning, compared to individual characteristics such as age, gender, educational level, self-efficacy, and social support.

Despite the novelty of the results, the following limitations warrant against overgeneralization. First, strictly taken, the cross-sectional design precludes causality, though, to estimate the direct and indirect effects of predictors on outcome variables, it was necessary to define causal models. Second, by default, it was impossible to review and retrieve the relevant and rapidly growing literature on the topic. To illustrate, in PubMed^®^, the term “COVID-19” yielded 139,380 hits (retrieved 31 May 2021), “COVID-19 and psychiatry” yielded 5632 hits; “COVID-19 and hospital staff” yielded 5314 hits; “COVID-19 and staff members and Iran” yielded 89 hits, “COVID-19 and staff members and review” yielded 902 hits, and “COVID-19 and staff and meta-analysis” yielded 46 hits. Given this, we are aware that the literature review must be incomplete by nature, which may have affected the formulation of hypotheses. In this regard, Asmundson and Taylor [81] critically commented on the following two points: First, with regard to the COVID-19 pandemic and the use of post-traumatic stress disorder (PTSD), they commented that much of the published work on PTSD related to the pandemic is flawed with respect adequate consideration of PTSD criteria, particularly the assessment of Criterion A, conceptualization of Criterion A events, use of outdated measures, and the timeframe of reported symptoms. As such, quality should still matter over quantity. Second, Asmundson and Taylor [81] also mentioned the issue of the so-called infodemic, which refers to the deluge of information about the pandemic, consisting of a mix of low- to high-quality information, and where also researchers are competing to contribute to the research on the psychology of COVID-19. Third, the survey was performed anonymously, though only participants willing and able and working full time on the COVID-19 front were enrolled. Given this, the sample characteristics might have biased the pattern of results. In particular, it is conceivable that full-time staff working frontline with sleep disturbances and mental health problems were particularly interested and motivated to participate in the study. Relatedly, fourth, it is conceivable that latent and unassessed dimensions, such as leisure time activities, physical activity patterns, substance use, medication use, irregular working schedules, sleep hygiene-related behavior, and pre-existing vulnerabilities, might have biased two or more dimensions in the same or opposite directions. Fifth, a control condition consisting of non-frontline or part-time working hospital staff would have yielded a more fine-grained picture of the associations between sleep and mental health.

Future studies should assess hospital staff members longitudinally; such an approach would allow to draw causal relationships between workplace-related demands and individual psychological dimensions. In this view, it would be important to see if interventions to improve sleep, such as cognitive-behavioral therapy for insomnia [71,82,83,84,85] or Acceptance and Commitment Therapy (ACT) [86], could beneficially impact on depression, anxiety, and stress among frontline hospital staff. Furthermore, future studies should also carefully assess the impact of the implementation of interprofessional learning curricula, as such curricula may enhance understanding of the work of other health professionals, which in turn could result in better patient care and satisfaction with medical services [87].

## 5. Conclusions

Compared to normative data of healthy controls, full-time frontline hospital staff members involved in the treatment of patients with COVID-19 reported higher insomnia and lower general health. Higher scores of depression, anxiety, and stress directly predicted higher insomnia and lower general health. Higher social support had a modest direct and indirect effect on insomnia and general health. Age, gender, educational level, and job experience were unrelated to insomnia and general health. The pattern of results suggests that irrespective of sociodemographic and job-related characteristics, interventions to improve depression, anxiety, stress may favorably impact on sleep and general health.

## Figures and Tables

**Table 1 ijerph-18-06289-t001:** Overview of sociodemographic and questionnaire-related descriptive data.

Variable		Total N (%)	Male N (%)	Female N (%)
Civil status	Single	122 (38)	35 (25.9)	87 (46.8)
	Married	187 (58.3)	95 (70.4)	92 (49.5)
	Divorced	12 (3.7)	5 (3.7)	7 (3.8)
	Total		135 (100)	186 (100)
Education	High school and diploma	18 (5.6)	10 (7.4)	8 (4.3)
	High diploma	57 (17.8)	24 (17.8)	33 (17.7)
	Bachelor	160 (49.8)	65 (48.1)	95 (51.1)
	Master and higher	86 (26.8)	36 (26.7)	50 (36.9)
	Total		135 (100)	186 (100)
Job experience (years)	< 1	15 (4.7)	4 (3)	11 (5.9)
	1–2	40 (12.5)	8 (5.9)	32 (17.2)
	3–5	64 (19.9)	16 (11.9)	48 (25.8)
	6–10	53 (16.5)	22 (16.3)	31 (16.7)
	≥11	149 (46.4)	85 (63.0)	64 (34.4)
	Total		135 (100)	186 (100)
	M (SD)		M (SD)	M (SD)
Age (years)	36.37 (7.72)		40.12 (7.32)	33.66 (6.82)
Stress-anxiety-depression inventory DASS-21	Depression	8.90 (5.09)	8.57 (5.31)	9.13 (4.93)
	Anxiety	9.09 (5.26)	8.92 (5.53)	9.22 (5.07)
	Stress	10.45 (4.91)	10.28 (5.08)	10.57 (4.80)
Total score DASS		28.44 (5.08)	27.77 (5.31)	28.92 (4.93)
General Self-Efficacy Scale-10 (GSES-10)		29.11 (5.72)	29.72 (6.09)	28.67 (5.41)
Multidimensional Scale of Social Support Questions (MSPSS)	Significant other subscale	19.66 (5.98)	19.58 (5.94)	19.73 (6.03)
	Family subscale	19.63 (5.87)	19.84 (5.66)	20.00 (6.03)
	Friends subscale	18.14 (5.77)	17.47 (6.00)	18.62 (5.57)
Total score MSPSS		57.74 (14.82)	56.90 (14.34)	58.34 (15.16)
General Health Questionnaire-12 (GHQ-12)	Negative symptoms of mental health	7.42 (4.43)	7.25 (4.64)	7.55 (4.28)
	Positive symptoms of mental health	8.22 (3.30)	8.23 (3.62)	8.22 (3.05)
Total score GHQ-12		18.80 (6.02)	18.98 (6.37)	18.67 (5.73)
Athens Insomnia Scale (AIS)	Sleep quantity and quality	7.43 (3.30)	7.57 (3.21)	7.33 (3.37)
	Daytime symptoms	4.58 (2.00)	4.64 (2.05)	4.53 (1.97)
Total score AIS		12.01 (4.65)	12.21 (4.55)	11.86 (4.73)

**Table 2 ijerph-18-06289-t002:** Overview of Pearson’s correlation coefficients between dimensions of stress, anxiety, depression, social support, self-efficacy, insomnia, and general health.

		**Dimensions**												
		**DASS**				**General Self-Efficacy**			**Social Support**					
DASS		2	3		4	5			6	7		8		9
1	Total score	0.91 ***	0.90 ***		0.90 ***	0.03			−0.45 ***	−0.39 ***		−0.40 ***		−0.33 ***
2	Depression	−	0.73 ***		0.75 ***	−0.04			−0.45 ***	−0.42 ***		−0.40 ***		−0.32 ***
3	Anxiety		−		0.72 ***	0.06			−0.40 ***	−0.34 ***		−0.38 ***		−0.29 ***
4	Stress				−	−0.05			−0.36 ***	−0.30 ***		−0.31 ***		−0.29 ***
5	General self-efficacy					−			0.27**	0.25 ***		0.25 ***		0.19 **
Social support														
6	Total score								−	0.90 ***		0.85 ***		0.78 ***
7	Significant others									−		0.71 ***		0.54 ***
8	Family											−		0.42 ***
9	Friends													−
		**Dimensions**												
		**General Health**						**Insomnia**						
DASS		10		11			12	13			14		15	
1	Total score	−0.43 ***		0.54 ***			−0.06	0.50 ***			0.45 ***		0.40 ***	
2	Depression	−0.49 ***		0.53 ***			−0.19 *	0.43 ***			0.40 ***		0.34 ***	
3	Anxiety	−0.30 ***		0.43 ***			−0.02	0.53 ***			0.48 ***		0.43 ***	
4	Stress	−0.39 ***		0.51 ***			−0.02	0.38 ***			0.34 ***		0.32 ***	
5	General self-efficacy	0.27 ***		−0.17 ***			0.34 ***	0.06			0.02		0.10 *	
Social support														
6	Total score	0.39 ***		−0.34 ***			0.26 ***	−0.29 ***			−0.27 ***		−0.24 ***	
7	Significant others	0.37 ***		−0.31 ***			0.26 ***	−0.24 ***			−0.21 **		−0.20 ***	
8	Family	0.27 ***		−0.22 ***			0.19 ***	−0.33 ***			−0.31 ***		−0.25 ***	
9	Friends	0.36 ***		−0.33 ***			0.20 **	−0.16 **			−0.15 **		−0.14**	
General health														
10	Total score	−		−0.84 ***			0.67 ***	−0.14 **			−0.16 **		0.−06	
11	Negative symptoms			−			−0.19 **	0.22 ***			0.21 ***		0.15 ***	
12	Positive symptoms						−	0.04			0.00		0.09	
Insomnia														
13	Total score							−			0.93 ***		0.79 ***	
14	Sleep quality and quantity										−		0.51 ***	
15	Daytime symptoms												−	

Notes: DASS = Depression, Anxiety, and Stress Scale; * = *p* < 0.05; ** = *p* < 0.01; *** = *p* < 0.001.

**Table 3 ijerph-18-06289-t003:** Equation models to predict insomnia. Model 1.

*r* _DASS-insomnia_	=	Direct Effect_dass_ β	+	Indirect Effect via Social Support β_DAS-social support_ × *r*_social support-insomnia_
*r* = 0.50	=	0.456	+	−0.45 × −0.087

Notes: DASS = Depression, Anxiety, and Stress Scale; overall score.

**Table 4 ijerph-18-06289-t004:** Equation models to predict insomnia. Model 2.

*r* _social support-insomnia_	=	Direct Effect_social support_ β	+	Indirect Effect via DASS β_DAS-insomnia_ × *r*_DAS,-insomnia_
*r* = −0.29	=	−0.087	+	−0.456 × −0.45

Notes: DASS = Depression, Anxiety, and Stress Scale; overall score.

**Table 5 ijerph-18-06289-t005:** Equation model to predict general health. Model 1.

*r* _DAS-general health_	=	Direct Effect_dass_ β	+	Indirect Effect via Social Support β_social support-general health_ × *r*_DASS-social support_
*r* = −0.43	=	−0.32	+	0.25 × −0.45

Notes: DASS = Depression, Anxiety, and Stress Scale; overall score.

**Table 6 ijerph-18-06289-t006:** Equation model to predict general health. Model 2.

*r* _social support-general health_	=	Direct Effect_social support_ β	+	Indirect Effect via DAS β_DAS-general health_ × *r*_DASS,-genral health_
*r* = 0.39	=	0.25	+	−0.32 × −0.45

Notes: DASS = Depression, Anxiety, and Stress Scale; overall score.

## Data Availability

Data are available upon request to competent experts.

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
