# Peer review of "Sources of Sleep Disturbances and Psychological Strain for Hospital Staff Working during the COVID-19 Pandemic"

_ijerph, 2021, doi:10.3390/ijerph18126289_

Round 1
Reviewer 1 Report
This is an interesting and well written study, conducted on a large group of participants. However, conclusions are quite generic and not innovative, but entire manuscript may be tempting for a large group of potential readers. According to my expertise, some minor criticism should be raised:
In the text, reference numbers should be placed in square brackets [ ] - this is consistent with MDPI manuscript preparation style, the same applies to References section
MethodsThere are no clear inclusion and exclusion criteria. Please elaborate briefly.
Discusion
These section is a little messy, mostly due to two study hypotheses and should be rearranged and clearly divided into two separate sections with conclusions at the end. Also, consider moving some paragraphs from Introduction here, as part of it reads like a Discussion e.g. L71-L106. Please follow MDPI publishing policy guidelines how to maintain a proper manuscript structure. Also, it would be convenient discussing some other works concerning Sars-Cov2 issues, as there is humongous amount of worldwide publications in this field going on right now. I feel that incorporating more references into Discussion section would add a lot of overall quality to the study. Yet, some factors mentioned in Table 2 might be cross-talked with psychological and systemic issues from other countries too e.g.https://www.mdpi.com/1660-4601/18/3/1281
Were there any limitations of the study? This is worth mentioning in a separate section, eg. 6. Study limitations
References
As mentioned above, adding and discussing a few more references, as there were similar reports from different parts of the world. This would greatly improve quality for potential readers. Adhering to MDPI formatting guidelines within this section is essential; please follow https://www.mdpi.com/journal/ijerph/instructions#preparation
Author Response
We thank Reviewer #1 for the care devoted to improve the quality of the manuscript. Please find the detailed point-by-point-response attached as a separate file. Thank you again for all your kind efforts.

Reviewer 2 Report
The authors report sleep disturbances in hospital staff members during the COVID-19 pandemic, with the working context as the major variable to impact individuals’ well-being. The study add further data on a overwhelming number of COVID related consequences on both general population and health workers.
Methodological comments:
The authors correctly stated that " In particular, it is conceivable that full-time staff working frontline with sleep disturbances and mental health problems were particularly interested and motivated to participate in the study", It would be informative the authors report the whole number of frontline hospital staff members at Farabi Hospital , and how this compare with their sample, was the sample representative of the whole staff members?
The statement "while the quality of the present results does not allow a deeper understanding of the underlying cognitive-emotional, neuronal and neurophysiological processes" is not necessary , considering the study design and methodology were not addressed to these processes
Author Response
We thank Reviewer #2 for the care devoted to improve the quality of the manuscript. Please find the detailed point-by-point-response attached as a separate file. Thank you again for all your kind efforts.

Round 2
Reviewer 1 Report
The entire manuscript was slightly improved, mostly due to other Reviewers' contribution. However, most of my remarks were not addressed correctly as authors claim in their response that 'they are misunderstanding' or they are having 'made good experiences, and we always got good feedback' whatever it means. Also, there are 11 (eleven authors) in this study, each one lazy enough not to rearrange Reference list, so in their response it goes as follows: ' this is often a question of taste and judgment, and this holds particularly true as regards the topic of the COVID-19 and its social, psychological, economical and health-related issues. The decision was to keeping the number of references unaltered, as any kind of reference up-date must have been remained arbitrary and incomplete by nature' - despite obvious language problems such an attitude towards reviewer is not acceptable.
To sum up: please, address my former remarks or I am going to withdraw the manuscript.
Author Response
Dear Reviewer #1, please find the detailed point-by-point-response attached as a separate file. Thank you for for your kind efforts.

This manuscript is a resubmission of an earlier submission. The following is a list of the peer review reports and author responses from that submission.